# Dietary Fiber Intake Influences Changes in Ankylosing Spondylitis Disease Status

**DOI:** 10.3390/jcm12041621

**Published:** 2023-02-17

**Authors:** Chuan Song, Lei Wang, Xiaojian Ji, Yiwen Wang, Lidong Hu, Xingkang Liu, Jiaxin Zhang, Simin Liao, Yishu Yan, Jian Zhu, Feng Huang

**Affiliations:** 1Department of Rheumatology and Immunology, The First Medical Center, Chinese PLA General Hospital, Beijing 100039, China; 2Medical School of Chinese PLA, Beijing 100039, China; 3Department of Gastroenterology, Air Force Medical Center, Air Force Medical University, Beijing 100142, China

**Keywords:** ankylosing spondylitis, functional bowel disorder, dietary fiber

## Abstract

The objectives of this study were to characterize dietary fiber (DF) intake in patients with ankylosing spondylitis (AS), to assess whether DF intake affects disease activity in AS, and to investigate the effect of DF intake on disease activity in AS in the context of functional bowel disease (FBD) symptoms. We recruited 165 patients with AS and divided them into two groups according to whether they had a high DF intake > 25 g/d to investigate the characteristics of people with high DF intake. Some 72 of the 165 AS patients (43%) met the criteria for high DF intake, which was more common in patients with negative FBD symptoms (68%). Data analysis revealed that DF intake was negatively associated with AS disease activity and did not differ statistically significantly from FBD symptoms. Multivariate adjusted models were used to explore the effect of DF intake on AS disease activity. ASDAS-CRP and BASDAI were stable and negatively correlated across models in both groups with and without FBD symptoms. Thus, DF intake positively affected disease activity in patients with AS. ASDAS-CRP and BASDAI were negatively correlated with DF intake.

## 1. Introduction

Ankylosing spondylitis (AS) is a long-standing, persistent condition characterized by systemic inflammation. It primarily affects the sacroiliac joint but can also impact other joints, tissues, and organs in the body. This chronic nature of the disease can significantly impact an individual’s quality of life, affecting their ability to perform daily activities and impacting their overall physical and mental wellbeing [1]. In patients with AS, as in the spondyloarthritis group, the link between joint and spinal symptoms and the gastrointestinal system has become an area of interest [2]. A systematic review of 30,410 patients with AS reported a 6.8% prevalence of inflammatory bowel disease (IBD) in the group, and that specific dietary pattern affected the disease activity of IBD [3,4]. Despite recognizing the influence of nutritional factors on IBD, there remains a gap in understanding the effect of dietary intake on gastrointestinal symptoms other than IBD, and its association with the disease activity in patients with AS.

Functional bowel disease (FBD) is a general term for disorders in which abdominal pain, constipation, diarrhea, and bloating or distention are the main symptoms [5]. Recent studies have suggested that improvement in FBD symptoms, particularly chronic diarrhea, may positively impact the disease state of individuals diagnosed with AS. This raises the possibility that addressing FBD symptoms may play a role in managing the overall condition of individuals with AS [6].

Diet patterns also alleviated joint symptoms, which may be related to gut microbiota. The gut microbiota play a crucial role in regulating the immune system and maintaining gut health, and alterations in the gut microbiome have been associated with the development of inflammatory diseases. Gut microbiota regulate bone growth, gastrointestinal permeability, and inflammation through various mechanisms [7,8]. Among the potential mechanisms, increased permeability between intestinal epithelial cells (IECs) contributes to the progression of AS [9]. In addition, gut microbiota promote new bone formation in AS by regulating the balance of helper T cells (Th17) and regulatory T cells (Treg) in the lamina propria (LP) to reduce the Th17/Treg ratio [10,11].

Despite this, little is known about the effect of dietary factors on AS disease activity. Ebringer and Wilson [2] confirmed the hypothesis that increased disease activity in AS patients may be associated with high-starch diets. One study noted that gastrointestinal symptoms might occur when consuming dairy products or flour [12]. Further research is needed to fully understand the effect of dietary factors on AS disease activity and the mechanisms underlying these effects. This information could have important implications for the management of AS and the development of targeted dietary interventions to improve patient outcomes.

Dietary fiber (DF) is a new term for residues in plant foods resistant to hydrolysis by human digestive enzymes, with its potential anti-inflammatory effects being the focus of current research [13]. These are thought to be related to dietary fiber’s effect on the gut microbiome. The gut microbiome is a complex community of microorganisms that inhabit the intestine and play a key role in maintaining health and preventing disease. Current data suggest that intestinal flora ferment DF in the colon and produce short-chain fatty acids (SCFAs), which reduce inflammation by inhibiting pro-inflammatory cytokines such as inducible nitric oxide synthase (iNOS), tumor necrosis factor (TNF)-α, interleukin (IL)-1b, IL-17a, IL-6, ans interferon (IFN)-γ and enhancing the expression of the anti-inflammatory cytokine IL-10 [14]. Therefore, studies establishing dietary patterns rich in DF and the risk of immunoinflammatory diseases such as rheumatoid arthritis (RA) have come into focus [15]. In Schönenberger’s [16] analysis, subjects eating the Mediterranean diet had significantly reduced pain from RA compared to those eating a regular diet. However, there is a paucity of research on AS and diet.

In this study, Our research objective is to analyze the level of DF intake in these patients and assess its association with AS disease activity. This examination could contribute to a better understanding of the role that diet plays in the management of AS, and inform the development of dietary interventions for individuals with this chronic inflammatory disease.

## 2. Material and Methods

This study was approved by the Ethics Committee of the PLA General Hospital (S2020-024-02). Participants in this study were 165 patients with AS that were consecutively recruited at the Department of Rheumatology and Immunology outpatient clinic, PLA General Hospital. All patients met the 1984 modified New York criteria for AS [17]. Exclusion criteria for patients with AS were: (i) age ≤ 18 years; (ii) cognitive impairment such that the patient could not cooperate in completing the questionnaire. All participants were given details of the study and signed an informed consent statement.

### 2.1. Clinical Features

Data collection was carried out by a team of trained investigators using a standardized questionnaire. The use of a structured questionnaire ensured that all necessary information was collected in a consistent and systematic manner and that the data gathered was of high quality. Data collected included demographic information on age, gender, body mass index (BMI); and disease-related characteristics, where disease-related characteristics were derived from the PLA clinical laboratory: human leucocyte antigen B-27 (HLA-B27), erythrocyte sedimentation rate (ESR), C-reactive protein (CRP) and functional bowel disease (FBD) symptoms. FBD symptoms were assessed in all participants using the Rome IV Diagnostic Questionnaire [5]. The smart phone Spondyloarthritis Management System (SpAMS) application was used to document the assessment of AS [18], including the Ankylosing Spondylitis Disease Activity Score based on C-reactive protein (ASDAS-CRP), the Bath Ankylosing Spondylitis Disease Activity Index (BASDAI), the Assessment of the Spondyloarthritis International Society Health Index (ASAS HI), the Bath Ankylosing Spondylitis Functional Index (BASFI), and the Bath Ankylosing Spondylitis Metrology Index (BASMI) [19,20,21,22,23]. Gut problems, including abdominal pain, constipation, diarrhea, and bloating had been investigated and reported previously [6].

### 2.2. DF Intake Standard

DF intake data were collected via the food frequency questionnaire (FFQ) over a year, including food type, quantity, and consumption frequency [24]. Food types include soluble dietary fiber-based foods (e.g., oats, legumes, mucilage) and insoluble dietary fiber-based foods (e.g., cellulose, lignin, wheat bran) [25]. The frequency of food and the amount of food intake included in the questionnaire was filled in by the patient’s own assessment (Appendix A). The data collection process involved the use of structured questionnaires, which were distributed to patients through electronic means. To ensure the accuracy and completeness of the collected data, the questionnaires were carefully reviewed and processed by trained and experienced dietitians. These professionals were responsible for verifying the data to ensure that they met the necessary quality standards and were suitable for analysis. The use of experienced dietitians in the data collection process helped to ensure that the results of this study were robust and could be relied upon to provide meaningful insights into the relationship between DF intake and AS disease activity. The DF content of each food item was estimated according to the China Dietary Guidelines [26]. Based on the data survey, a statistical analysis was conducted to evaluate the distribution of DF intake among the participants. The results revealed that the distribution of DF intake was highly skewed. As a result, the subjects were divided into two distinct categories, namely high and low DF intake, to facilitate further analysis and interpretation of the data. According to the European Prospective Investigation into Cancer and Nutrition (EPIC) by the European Food Safety Authority (EFSA), a DF of >25 g/d in adults has a positive anti-inflammatory effect and is considered “high” [27].

### 2.3. Statistics

R 4.2.0 (www.R-project.org, 26 June 2022) was used to analyze data. Continuous variables were expressed as mean ± standard deviation (normal distribution) or median and quartiles (skewed distribution), and categorical variables were described in frequency or percentage. A Kruskal–Wallis (skewed distribution) test and Chi-square test (categorical variable) were used to determine statistical differences between the two groups. A linear regression model assessed the correlation between dietary intake and clinical data indicators. Both the unadjusted model and the multivariable-adjusted model are reported. The following variables were included in the covariate screening: age, sex, body mass index, HLA-B27, smoking status, ESR, CRP, history of tumor necrosis factor inhibitor (TNFi) use, history of non-steroidal anti-inflammatory drug (NSAID) use, history of Chinese medicine (CM) use, history of conventional synthetic disease-modifying antirheumatic drug (csDMARD) use, past history of psoriatic arthritis (PsA), and past history of uveitis. Among them, gender and age were fixed adjustment variables. Results from the unadjusted, minimally adjusted, and fully adjusted analyses are shown simultaneously, as suggested by the strengthening of the reporting of observational studies in epidemiology (STROBE) statement [28]. The criterion for model I was a *p*-value of <0.10 for the regression coefficient of the covariate on the outcome variable, and the standard for model II was a change of more than 10% in the regression coefficient of the risk factor resulting from the introduction of the covariate in the basic model [29]. All tests were two-tailed, and *p* ≤0.05 was considered statistically significant.

## 3. Result

### 3.1. Patient Characteristics

The characteristics of the patients in this cross-sectional study are shown in Table 1. Of the 165 patients, 11 were diagnosed with colonic ulcers (CU), including five cases of ulcerative colitis, three cases of Crohn’s disease, two cases of autoimmune-associated colitis, and one case of NSAID-associated gastrointestinal ulcer. Overall, 43 percent of patients met the criteria of high DF intake. Based on the Chi-square test, DF intake had no statistically significant effect on FBD (*p* = 0.18). Levels of CRP (*p* < 0.001) and ESR (*p* < 0.001) were lower in the high DF group. At the same time, the high DF group had lower ASDAS-CRP (*p* < 0.001), BASDAI (*p* < 0.01), ASAS HI (*p* = 0.002), and BASFI (*p* < 0.001) scores. No statistical difference was observed between the two DF groups in BASMI (*p* = 0.81). In the intestinal disease survey, only 36% (26/72) of patients in the high DF group exhibited gastrointestinal symptoms. In comparison, 49% (46/93) of patients in the low DF group had the same gastrointestinal symptoms. Details on effect sizes and confidence intervals for the data are provided in Appendix A.

### 3.2. DF Intake with Disease Activity

In prior research, adjusted regression models have accounted for the impact of FBD symptoms on the disease activity of patients with AS. To control for this effect, FBD symptoms were incorporated as a stratifying variable in the regression model. This adjustment is critical to isolate and comprehend the specific relationship between DF intake and AS disease activity, and to mitigate the potential confounding variables’ impact on the results of the analysis. In the unadjusted model, the high DF group was negatively correlated with ASDAS-CRP *(β* = −0.8, 95% CI: −1.1, −0.6; *p* < 0.01), BASDAI (*β* = −0.9, 95% CI: −1.3, −0.5; *p* < 0.01), ASAS HI (β = −1.4 95% CI: −2.5, −0.4; *p* < 0.01), and BASFI (*β* = −0.7, 95 %CI: −1.1, −0.3; *p* < 0.01), which was consistent with single-factor results. In the adjusted model I, the effects of the high DF group on ASDAS-CRP *(β* = −0.3, 95% CI: −0.5, −0.2; *p* < 0.01) remained significant. BASDAI (*β =* −0.8, 95% CI: −1.6, 0.0; *p* = 0.06) showed marginal significance in patients with positive FBD symptoms, which a slightly smaller sample size may cause. ASAS HI (*β* = −1.9, 95% CI: −3.6, −0.2; *p* = 0.03) was statistically significant in patients with positive FBD symptoms, while BASMI (*β* = 0.5, 95%CI: −0.1, 1.1; *p* = 0.08) and BASFI (*β* = −0.4, 95% CI: −0.9, 0.0; *p* = 0.07) were not statistically significant in this model. In adjusted model II, ASDAS-CRP (*β* = −0.3, 95% CI: −0.5, −0.2; *p* < 0.01) and BASDAI (*β* = −0.8, 95% CI: −1.3, −0.4; *p* < 0.01) showed significant effects, ASAS-HI (*β* = −2.1, 95% CI: −4.0, −0.1; *p* = 0.04) and BASFI (*β* = −1.0, 95% CI: −1.7, −0.3; *p* < 0.01) showed statistical differences among patients with positive FBD symptoms, while BASMI (*β* = 0.5, 95% CI: −0.1, 1.1; *p* = 0.08) showed no statistical significance under this model (Table 2 and Table 3 and Appendix A).

## 4. Discussion

The purpose of the present study was to examine the correlation between daily fiber intake and the level of disease activity in individuals diagnosed with AS. The results indicated a strong association between high fiber consumption and reduced disease activity in patients with AS. The data were analyzed, controlling for various factors, and the findings revealed a persistent negative correlation between high fiber intake and both ASDAS-CRP and BASDAI scores. These findings reinforce and add to the existing body of literature that suggests the positive impact of fiber intake on AS. The results of this study can be valuable for healthcare professionals in promoting healthy dietary choices for individuals with AS, and for future research in this field. 

DF is essential for gut health and has been shown to relieve gastrointestinal symptoms through its anti-inflammatory effects [30,31,32]. Interestingly, high DF group intake was not correlated with patients’ gastrointestinal symptoms in this study. This may have several causes. First, patients had different gastrointestinal symptoms; about 43 patients had diarrhea as their primary symptom, and the rest had other symptoms. In addition, patients had different types of DF intake. According to our dietary survey (the FFQ), the DF sources for AS patients in this study were primarily low-FODMAP brown rice, quinoa, and oats, which were beneficial for diarrhea patients [33,34]. Second, it is important to consider that not all types of DF may have the same impact on both gastrointestinal symptoms and anti-inflammatory effects. Different types of DF have unique physical and chemical properties that can impact the digestive system differently. Insoluble DF (e.g., cellulose) can increase the transport rate of undigested food and ferment in the colon, leading to gastrointestinal symptoms of bloating and constipation, but not for diarrhea or abdominal pain patients [35]. As a result, it is possible that the type of DF consumed may have a different impact on both gastrointestinal symptoms and anti-inflammatory effects. Inflammation is a key factor in the development and progression of FBD and AS, and the anti-inflammatory properties of DF may have a significant impact on reducing inflammation levels in the body. Therefore, although this study found no statistical difference in the effect of DF intake on gastrointestinal symptoms, its anti-inflammatory effect could still improve FBD disease activity in patients with AS.

In the regression equations of different models, we find that ASDAS-CRP and BASDAI are strongly associated with DF intake after adjusting for covariates. Of most significant importance is that inflammation has been identified as a major proponent of metabolic disorders [36]. The anti-inflammatory effect of DF may be mediated through its impact on the intestinal flora. The mechanism may be to promote the growth of beneficial intestinal bacteria, reduce the abundance of pro-inflammatory intestinal microorganisms, and affect the production of metabolites that regulate inflammation. It has been shown that fructooligosaccharides (FOSs) in quinoa changed the proportion of *Bacteroidetes* and *Firmicutes*, with a significant increase in SCFAs in the cecum of rats and mice [37,38]. It is possible that the ratio of *Bacteroidetes* and *Firmicutes* in the intestinal flora of AS patients with high DF intake was changed to be more beneficial. A study by Qin et al. (2010) found that high DF intake was associated with an increase in the proportion of Bacteroidetes and a decrease in the proportion of Firmicutes in the gut microbiome [39]. Bacteroidetes have been associated with improved gut health and decreased inflammation, while Firmicutes have been linked to obesity and metabolic disorders [40]. After dietary fiber intake, several SCFAs-producers significantly increase, including *Lachnospira*, *Akkermansia*, *Bifidobacterium*, *Lactobacillus*, *Ruminococcus*, *Roseburia*, *Clostridium*, *Faecalibacterium*, and *Dorea* [41,42,43]. Among them, *Akkermansia* and *Lachnospira* restore intestinal permeability, reduce the secretion of TNF-α, increase IL-10 production, and act as anti-inflammatory agents [44,45,46].

The intestinal flora produces SCFAs, mainly acetate, propionate, and butyrate, present in intestinal mucosa and feces in a molar ratio of approximately 3:1:1 from food components [47]. Gene expression analysis of ileal samples from AS patients showed a decrease in intestinal tight junction (TJ) protein expression, leading to increased intestinal vascular barrier permeability and intestinal and joint inflammation [48]. In contrast, acetate maintains intestinal homeostasis and protects the intestine from inflammation by modulating intestinal immunoglobulin A (IgA) [49]. Butyrate can reduce the production of lipopolysaccharide (LPS)-stimulated neutrophils, cytokine-induced neutrophil chemoattractant (CINC)-2αβ, TNF-α, and NO [50], and inhibit the phagocytosis and killing function essential for neutrophilic inflammation in the gut of AS patients [49,51]. However, an animal study [14] on the relationship between HLA-B27 gene background and fecal SCFAs suggests that the fecal SCFAs concentration in HLA-B27 transgenic rats differs from that of wild-type rats. Fecal butyric acid and valeric acid concentrations were higher than those of wild-type rats at 6 and 16 weeks, and intestinal SCFAs receptor expression was significantly changed, with free fatty acid receptor 2 (FFAR2) and niacin receptor subtype 1 (NIACR1) significantly upregulated, and FFAR3 significantly downregulated. In addition, propionic acid interfered with HLA-B27 rats to dramatically reduce inflammation levels, including IL-1β, IL-17A, and INF-γ. This change was not achieved by the induction of forkhead box (Fox)P3+ T cells, whose mRNA expression of IL-1β, IL-17A, and INF-γ was not increased; in HLA-B27 rats intervened with using butyric acid, the alteration in inflammation levels was weaker than the effect of propionate intervention. Since drug (e.g., NSAID, TNFi, etc.) use in human studies may have altered gut microbes’ metabolism, human studies’ results differ from animal studies. Based on the available results, we hypothesize that supplementation of DF and the associated increase in propionic acid production may further reduce the level of inflammation in the body, or increase the anti-inflammatory effect of the drug; further experiments are required to confirm this. These experiments may involve controlled trials that examine the impact of DF supplementation on inflammation levels and compare the results to those obtained from placebo or other interventions. Further research is necessary to confirm this relationship and fully understand the potential impact of DF supplementation on the body.

### Limitations

The current study has several limitations that must be acknowledged. Firstly, the use of the FFQ to assess fiber intake may have introduced limitations in terms of accuracy due to the subjective nature of self-reported dietary data and the recall bias of the participants. This hinders our ability to accurately determine the exact DF intake and type for each individual, which is crucial for developing targeted and effective dietary interventions. Secondly, it is important to note that the study was designed as an observational trial, which provides valuable insights but is not sufficient to establish causality between DF intake and the reduction of inflammation in individuals with AS. Further studies, such as randomized controlled trials, are necessary to establish a causal relationship and provide robust evidence of the anti-inflammatory effects of fiber. Finally, it is also possible that other dietary factors or habits, such as a high fermentation diet that has been linked to reducing inflammation, may have influenced the results of this study [52].

## 5. Conclusions

DF intake has a positive effect on disease activity in patients with AS. High DF intake is associated with low ASDAS-CRP and BASDAI scores.

## Figures and Tables

**Table 1 jcm-12-01621-t001:** Baseline characteristics of participants (*n* =165).

Dietary Fiber Intake			
Characteristic	High (*n* = 72)	Low (*n* = 93)	*p*
Age (y)	34.29 ± 8.23	33.58 ± 9.07	0.32
Female, *n* (%)	12 (16.67%)	15 (16.13%)	0.93
BMI <= 25 (kg/m^2^)	48 (66.7%)	48 (51.6%)	0.05
HLA-B27 +, *n* (%)	61 (85.92%)	81 (88.04%)	0.69
CRP (mg/L)	1.4 (0.6–3.9)	8.9 (2.2–16.4)	<0.001
ESR (mm/h)	6.0 (2.2–10.8)	13.0 (6.0–31.0)	<0.001
BASDAI	1.2 ± 1.1	2.3 ± 1.5	<0.001
ASDAS-CRP	1.2 ± 0.6	2.1 ± 0.9	<0.001
ASAS.HI	2.7 ± 3.2	4.4 ± 3.8	0.002
BASMI	1.2 ± 1.8	1.1 ± 1.6	0.81
Bowel disease			
FBD, + *n* (%)	23 (33.3%)	37 (44.0%)	0.18
CU, + *n* (%)	3 (4.2%)	8 (8.6%)	0.26
Medications, + *n* (%)			
NSAID	53 (73.6%)	80 (86.0%)	0.05
DMARD			
TNFi	25 (34.7%)	28 (30.1%)	0.52
csDMARD	18 (25.0%)	18 (19.4%)	0.38
CM	31 (43.1%)	40 (43.0%)	0.10

Unless otherwise specified, values are shown as mean ± SD or median (IQR). Missing data, *n* (%): HLA-B27 2 (1.2%), CRP 8 (4.8%), ESR 10 (6.0%), ASDAS-CRP 8 (4.8%), BASMI 10 (6.0%), FBD 12 (7.2%). BMI: body mass index; HLA-B27: human leucocyte antigen B-27; CRP: C-reactive protein; ESR: erythrocyte sedimentation rate; ASDAS-CRP: Ankylosing Spondylitis Disease Activity Score based on CRP; BASDAI: Bath Ankylosing Spondylitis Disease Activity Index; ASAS HI: Assessment of Spondyloarthritis International Society Health Index; BASFI: Bath Ankylosing Spondylitis Functional Index; BASMI: Bath Ankylosing Spondylitis Metrology Index; FBD: functional bowel disorder; CU: colonic ulcer; NSAID: non-steroidal anti-inflammatory drug; DMARD: disease-modifying antirheumatic drug; TNFi: tumor necrosis factor inhibitor; csDMARD: conventional synthetic DMARD; CM: Chinese medicine; SCFAs: short-chain fatty acids; SD: standard deviation; IQR: interquartile range.

**Table 2 jcm-12-01621-t002:** Relationship between dietary fiber intake and ASDAS-CRP.

Dietary Fiber Intake				
ASDAS-CRP	Low	High		
Crude Model		β	95% CI	*p*-Value
FBD symptoms (+)	Ref	−1.0	(−1.5, −0.5)	<0.01
FBD symptoms (-)	Ref	−0.8	(−1.0, −0.5)	<0.01
Total	Ref	−0.9	(−1.1, −0.6)	<0.01
**Model I**				
FBD symptoms (+)	Ref	−0.3	(−0.5, 0.0)	0.01
FBD symptoms (−)	Ref	−0.4	(−0.6, −0.2)	<0.01
Total	Ref	−0.3	(−0.5, −0.2)	<0.01
**Model II**				
FBD symptoms (+)	Ref	−0.4	(−0.7, −0.1)	0.01
FBD symptoms (−)	Ref	−0.3	(−0.6, −0.1)	<0.01
Total	Ref	−0.3	(−0.5, −0.2)	<0.01

Model I adjusted for sex; age; lnESR, lnCRP. Model II adjusted for: sex; age CM-use; NSAID-use; BMI; lnESR; lnCRP. Missing data, *n* (%): CRP 8 (4.8%), ESR 10 (6.0%), ASDAS-CRP 8 (4.8%), FBD 12 (7.2%). BMI: body mass index; CRP: C-reactive protein; ESR: erythrocyte sedimentation rate; ASDAS-CRP: Ankylosing Spondylitis Disease Activity Score based on CRP; FBD: functional bowel disorder; NSAID: non-steroidal anti-inflammatory drug; CM: Chinese medicine.

**Table 3 jcm-12-01621-t003:** Relationship between dietary fiber intake and BASDAI.

Dietary Fiber Intake				
BASDAI	Low	High		
Crude Model		β	95% CI	*p*-Value
FBD symptoms (+)	Ref	−1.2	(−2.0, −0.4)	<0.01
FBD symptoms (−)	Ref	−0.7	(−1.1, −0.2)	<0.01
Total	Ref	−0.9	(−1.3, −0.5)	<0.01
**Model I**				
FBD symptoms (+)	Ref	−0.8	(−1.6, 0.0)	0.06
FBD symptoms (−)	Ref	−0.8	(−1.3, −0.3)	<0.01
Total	Ref	−0.8	(−1.2, −0.4)	<0.01
**Model II**				
FBD symptoms (+)	Ref	−1.1	(−1.9, −0.3)	0.01
FBD symptoms (−)	Ref	−0.7	(−1.2, −0.3)	<0.01
Total	Ref	−0.8	(−1.3, −0.4)	<0.01

Model I adjusted for: sex; age; lnESR; lnCRP. Model II adjusted for: sex; age CM-use; NSAID-use; BMI; lnESR; lnCRP. Missing data, *n* (%): CRP 8 (4.8%), ESR 10 (6.0%), FBD 12 (7.2%). BMI: body mass index; CRP: C-reactive protein; ESR: erythrocyte sedimentation rate; BASDAI: Bath Ankylosing Spondylitis Disease Activity Index; FBD: functional bowel disorder; NSAID: non-steroidal anti-inflammatory drug; CM: Chinese medicine.

## Data Availability

The data presented in this study are available on request from the corresponding author.

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
