# Peer review of "Dietary Fiber Intake Influences Changes in Ankylosing Spondylitis Disease Status"

_jcm, 2023, doi:10.3390/jcm12041621_

Round 1
Reviewer 1 Report
Song et al have undertaken an important study to shed light on the influence of dietary fiber intake on ankylosing spondylitis.
Comments
Authors give a clear background and articulate gaps in knowledge their study sought to address.
Methods
Could the authors please rename subsection 2.2. “High DF intake standard” to reflect the intention of this subsection? For example, “Definition of DF intake”. Under this subsection, it is mentioned that the decision to categorise participants into low and high DF intake was based on a data survey. Which data survey?
Authors cite a review on design, validation and utilisation of FFQs which does not tell the reader the exact questionnaire authors used to measure DF intake. Could you please also cite the FFQ you used or preferably attached a copy of it as a supplement?
Reference 25 “Khan, I.A.; Yiqun, C.; Zongshuai, Z.; Ijaz, M.U.; Brohi, S.A.; Ahmad, M.I.; Shi, C.; Hussain, M.; Huang, J.; Huang, M. Occurrence 303 of Heterocyclic Amines in Commercial Fast-Food Meat Products Available on the Chinese Market and Assessment of Human 304 Exposure to These Compounds. J. Food Sci. 2019, 84, 192–200, doi:10.1111/1750-3841.14418.” does not show Chinese Dietary guidelines. Could the authors please cite an appropriate reference?
Could the authors add a sample size calculation?
Have the authors adjusted for comorbidities that may influence AS severity?
Chi-square test is used for two variables i.e., unadjusted analysis and should not be used to make inference.
Could you clarify a prior what was included in the full model? Looking at your model1 and model2, it appears model 2 does not fully build on model 1 for example it leaves out sex and age? Could you please explain why? Because your fully adjusted model does not account for the age and sex, it is difficult to comment on your results as age should really be adjusted for in addition to other important confounders.
Results
“DF intake had no statistically significant effect on FBD (P = 0.18)” please add that this is from chi-square test.
Remove this sentence: “In the adjusted regression model, in previous studies, FBD symptoms were included as a covariate due to the influence of FBD symptoms on the disease activity of AS patients”, from your results. Doesn’t belong.
Unadjusted results have extensively been presented in the text and discussed which really is misleading. I would urge author to focus on adjusted results acknowledging that there might be residual confounding given not many factors were controlled for.
Author Response
Point 1: Could the authors please rename subsection 2.2. “High DF intake standard” to reflect the intention of this subsection? For example, “Definition of DF intake”. Under this subsection, it is mentioned that the decision to categorise participants into low and high DF intake was based on a data survey. Which data survey?
Response 1: Dear Reviewer, The title of the second subsection has been revised based on your comments. The phrase "this data survey" refers to the questionnaire conducted for this study.
Point 2: Authors cite a review on design, validation and utilisation of FFQs which does not tell the reader the exact questionnaire authors used to measure DF intake. Could you please also cite the FFQ you used or preferably attached a copy of it as a supplement?
Response 2: A copy of the FFQ questionnaire has been added to the attached zip file for your review.
Point 3: Reference 25 “Khan, I.A.; Yiqun, C.; Zongshuai, Z.; Ijaz, M.U.; Brohi, S.A.; Ahmad, M.I.; Shi, C.; Hussain, M.; Huang, J.; Huang, M. Occurrence 303 of Heterocyclic Amines in Commercial Fast-Food Meat Products Available on the Chinese Market and Assessment of Human 304 Exposure to These Compounds. J. Food Sci. 2019, 84, 192–200, doi:10.1111/1750-3841.14418.” does not show Chinese Dietary guidelines. Could the authors please cite an appropriate reference?
Response 3: The references have been revised as you requested
Wang, S.-S.; Lay, S.; Yu, H.-N.; Shen, S.-R. Dietary Guidelines for Chinese Residents (2016): Comments and Comparisons. J Zhejiang Univ Sci B 2016, 17, 649–656, doi:10.1631/jzus.B1600341.
Point 4: Could the authors add a sample size calculation?
Response 4: Dear Reviewer, The sample size was not calculated. One reason is that no relevant studies were found for reference. The second is that there has been a statistically significant p-value change in the current sample size under different variable adjustments.
Point 5: Have the authors adjusted for comorbidities that may influence AS severity?
Response 5: In the process of screening covariates, it was found that comorbidities of ankylosing spondylitis were not sufficient for covariate requirements, and in the revised manuscript, I clarified this issue.
Point 6: Chi-square test is used for two variables i.e., unadjusted analysis and should not be used to make inference.
Response 6: Dear reviewers, the unadjusted variable model is only presented to the reader as a preliminary result and is not used for inference, the focus is still on model 1 and model 2. I may have misunderstood you because of my lack of detail in writing, and I have addressed this issue in the revised manuscript.
Point 7: Could you clarify a prior what was included in the full model? Looking at your model1 and model2, it appears model 2 does not fully build on model 1 for example it leaves out sex and age? Could you please explain why? Because your fully adjusted model does not account for the age and sex, it is difficult to comment on your results as age should really be adjusted for in addition to other important confounders.
Response 7: Dear Reviewer, In response to your question I am attaching the code for covariate screening. Model 1 and Model 2 are based on inherently adjusted variables, and different covariates are screened out and modeled according to different criteria, respectively. We have reworked the data according to your suggestion and the modified parts are presented in the manuscript for your review.
Sys.setlocale(category = 'LC_ALL', locale = 'English_United States.1252');
.libPaths(file.path(R.home(),'library'));
library(doBy);
options(timeout=600);
library(plotrix);
library(stringi);
library(stringr);
library(survival);
library(rms);
library(nnet);
library(car);
library(mgcv);
pdfwd<-6; pdfht<-6;
load('D:/EmpowerRCH/Analysis/hff/cleandata2.Rdata');
if (length(which(ls()=='EmpowerStatsR'))==0) EmpowerStatsR<-get(ls()[1]);
names(EmpowerStatsR)<-toupper(names(EmpowerStatsR));
originalVNAME<-names(EmpowerStatsR);
ofname<-'hff_13_tbl';
attach(EmpowerStatsR);
sink(paste(ofname,'_datastep.lst',sep=''));
print('Creating new variable: ASAS.HI.CONT');
ASAS.HI.CONT<- ASAS.HI;
summary(ASAS.HI.CONT);
EmpowerStatsR<-cbind(EmpowerStatsR,ASAS.HI.CONT);
print('Creating new variable: BASMI.CONT');
BASMI.CONT<- BASMI;
summary(BASMI.CONT);
EmpowerStatsR<-cbind(EmpowerStatsR,BASMI.CONT);
rm(ASAS.HI.CONT,BASMI.CONT);
sink();
vname<-c(NA,'SEX','SEX.0','SEX.1','HIGH.FIBER.DIET','HIGH.FIBER.DIET.0','HIGH.FIBER.DIET.1','AGE','BMI','HLA.B27','HLA.B27.0','HLA.B27.1','SMOKE','SMOKE.0','SMOKE.1','BASDAI','ASDAS','ESR','CRP','ASAS.HI','ASAS.HI.0','ASAS.HI.1','ASAS.HI.2','ASAS.HI.3','ASAS.HI.4','ASAS.HI.5','ASAS.HI.6','ASAS.HI.7','ASAS.HI.8','ASAS.HI.9','ASAS.HI.10','ASAS.HI.11','ASAS.HI.13','ASAS.HI.16','ASAS.HI.17','BASFI','BASMI','BASMI.0','BASMI.1','BASMI.2','BASMI.3','BASMI.4','BASMI.5','BASMI.6','BASMI.7','BASMI.9','HPSA','HPSA.0','HPSA.1','UVEITIS','UVEITIS.0','UVEITIS.1','HNSAID','HNSAID.0','HNSAID.1','HTNFI','HTNFI.0','HTNFI.1','HCM','HCM.0','HCM.1','ASAS.HI.CONT','BASMI.CONT')[-1];
vlabel<-c(NA,'SEX',' 0',' 1','HIGH.FIBER.DIET',' 0',' 1','AGE','BMI','HLA.B27',' 0',' 1','SMOKE',' 0',' 1','BASDAI','ASDAS','ESR','CRP','ASAS.HI',' 0',' 1',' 2',' 3',' 4',' 5',' 6',' 7',' 8',' 9',' 10',' 11',' 13',' 16',' 17','BASFI','BASMI',' 0',' 1',' 2',' 3',' 4',' 5',' 6',' 7',' 9','HPSA',' 0',' 1','UVEITIS',' 0',' 1','HNSAID',' 0',' 1','HTNFI',' 0',' 1','HCM',' 0',' 1','ASAS.HI continuous','BASMI continuous')[-1];
varused4this <- c('SEX','HIGH.FIBER.DIET','AGE','BMI','HLA.B27','SMOKE','BASDAI','ASDAS','ESR','CRP','ASAS.HI','BASFI','BASMI','HPSA','UVEITIS','HNSAID','HTNFI','HCM','ASAS.HI.CONT','BASMI.CONT');
pkgs<-c('mgcv','geepack','fmsb');
for (g in pkgs) {
if (!(g %in% rownames(installed.packages()))) install.packages(g,repos='https://cloud.r-project.org');
}
library(mgcv);
library(geepack);
library(fmsb);
WD <- EmpowerStatsR; rm(EmpowerStatsR); gc();
title<-'协变量检查与筛选';
wd.subset='';
weights.var<- NA;
yvname<-c('ASAS.HI.CONT','BASMI.CONT','BASFI','ASDAS','BASDAI');
ydist<-c('gaussian','gaussian','gaussian','gaussian','gaussian');
ylink<-c('identity','identity','identity','identity','identity');
ylv<-c(0,0,0,0,0);
svname<-c('BMI','HLA.B27','SMOKE','ESR','CRP','HTNFI','HNSAID','HPSA','UVEITIS','HCM');
sdf<-c(0,0,0,0,0,0,0,0,0,0);
slv<-c(0,2,2,0,0,2,2,2,2,2);
par1<-10;
colvname<- NA;
xvname<-c('HIGH.FIBER.DIET');
xlv<-c(2);
cox<- 0;
timevar<- NA;
vname.start<- NA;
avname<-c('SEX','AGE');
saf<-c(0,0);
alv<-c(2,0);
subjvname<- NA;
gee.TYPE<-NA;
dec<-4;
##R package## mgcv geepack fmsb ##R package##;
pvformat<-function(p,dec) {
pp <- sprintf(paste("%.",dec,"f",sep=""),as.numeric(p))
if (is.matrix(p)) {pp<-matrix(pp, nrow=nrow(p)); colnames(pp)<-colnames(p);rownames(pp)<-rownames(p);}
lw <- paste("<",substr("0.00000000000",1,dec+1),"1",sep="");
pp[as.numeric(p)<(1/10^dec)]<-lw
return(pp)
}
numfmt<-function(p,dec) {
if (is.list(p)) p<-as.matrix(p)
pp <- sprintf(paste("%.",dec,"f",sep=""),as.numeric(p))
if (is.matrix(p)) {pp<-matrix(pp, nrow=nrow(p));colnames(pp)<-colnames(p);rownames(pp)<-rownames(p);}
pp[as.numeric(p)>10000000]<- "inf."
pp[is.na(p) | gsub(" ","",p)==""]<- ""
pp[p=="-Inf"]<-"-Inf"
pp[p=="Inf"]<-"Inf"
return(pp)
}
vifSelect<-function(dfr, t=5, fix=NA, weights=1){
if(!class(dfr)[1]=="data.frame") dfr<-data.frame(dfr)
vnames<-names(dfr); nvar<-length(vnames);
if (!is.na(fix[1])) {v4slt<-(1:nvar)[-fix];} else {v4slt<-(1:nvar);}
vifmat<-matrix(NA,nrow=nvar,ncol=1)
vifmax<-100; v.rm<-0; k<-1
while (vifmax>=t) {
vnum <-(0:nvar)[-match(v.rm,(0:nvar))]
for (v in vnum) {
fml <- paste(vnames[v],"~",paste(vnames[vnum[vnum!=v]],collapse="+"))
lm.mdl<-summary(lm(formula(fml),weights=weights, data=dfr));
print(fml);print(lm.mdl)
vifmat[v,k]<-round(1/(1-lm.mdl$r.squared),1)
}
vifmax<-max(as.numeric(vifmat[v4slt,k]),na.rm=TRUE)
if (vifmax<t) break
v.rm<-c(v.rm,v4slt[which(as.numeric(vifmat[v4slt,k])==vifmax)[1]])
k<-k+1; vifmat<-cbind(vifmat,NA);
}
slt.vv<-!is.na(vifmat[v4slt,k])
vif.Rm<-vnames[is.na(vifmat[,k])]
vifmat2 <- rbind(c(" ",paste("Step",(1:k))),cbind(vnames,vifmat))
dfcmpt<-dfr[apply(is.na(dfr),1,sum)==0,]
dfvmax<-apply(dfcmpt,2,max); dfvmin<-apply(dfcmpt,2,min); singlar<-(dfvmax==dfvmin);
if (sum(singlar)>0) {
slt.vv<-(!is.na(vifmat[,k]) & (!singlar))[v4slt]
vif.Rm<-vnames[is.na(vifmat[,k]) | singlar]
vifmat2 <-cbind(vifmat2, c("Singlarity check", singlar))
}
return(list(slt.vv,vif.Rm,vifmat2))
}
mat2htmltable<-function(mat) {
t1<- apply(mat,1,function(z) paste(z,collapse="</td><td>"))
t2<- paste("<tr><td>",t1,"</td></tr>")
return(paste(t2,collapse=" "))
}
chkMatrixSinglar <- function(mat) {
oo<-rep(0,ncol(mat))
tmp<-mat[apply(is.na(mat),1,sum)==0,]
oo[apply(tmp,2,function(z) length(levels(as.factor(z)))==1)]<-1
rr<-cor(tmp)
for (i in (2:ncol(rr))) if (sum(rr[1:(i-1),i]>0.999)>0) oo[i]<-1
if (sum(oo)>0) print(rr)
return(oo)
}
setgam<-function(fml,yi) {
if (ydist[yi]=="") ydist[yi]<-"gaussian"
if (ydist[yi]=="exact") ydist[yi]<-"binomial"
if (ydist[yi]=="breslow") ydist[yi]<-"binomial"
if (ydist[yi]=="gaussian") mdl<-gam(formula(fml),weights=wdtmp$weights,data=wdtmp, family=gaussian(link="identity"))
if (ydist[yi]=="binomial") mdl<-gam(formula(fml),weights=wdtmp$weights,data=wdtmp, family=binomial(link="logit"))
if (ydist[yi]=="poisson") mdl<-gam(formula(fml),weights=wdtmp$weights,data=wdtmp, family=poisson(link="log"))
if (ydist[yi]=="gamma") mdl<-gam(formula(fml),weights=wdtmp$weights,data=wdtmp, family=Gamma(link="inverse"))
if (ydist[yi]=="negbin") mdl<-gam(formula(fml),weights=wdtmp$weights,data=wdtmp, family=negbin(c(1,10), link="log"))
return(mdl)
}
setgee<-function(fml,yi) {
if (ydist[yi]=="") ydist[yi]<-"gaussian"
if (ydist[yi]=="exact") ydist[yi]<-"binomial"
if (ydist[yi]=="breslow") ydist[yi]<-"binomial"
if (ydist[yi]=="gaussian") md<-try(geeglm(formula(fml),id=wdtmp[,subjvname],corstr=gee.TYPE,family="gaussian",weights=wdtmp$weights,data=wdtmp))
if (ydist[yi]=="binomial") md<-try(geeglm(formula(fml),id=wdtmp[,subjvname],corstr=gee.TYPE,family="binomial",weights=wdtmp$weights,data=wdtmp))
if (ydist[yi]=="poisson") md<-try(geeglm(formula(fml),id=wdtmp[,subjvname],corstr=gee.TYPE,family="poisson",weights=wdtmp$weights,data=wdtmp))
if (ydist[yi]=="gamma") md<-try(geeglm(formula(fml),id=wdtmp[,subjvname],corstr=gee.TYPE,family="Gamma",weights=wdtmp$weights,data=wdtmp))
if (ydist[yi]=="negbin") md<-try(geeglm.nb(formula(fml),id=wdtmp[,subjvname],corstr=gee.TYPE,weights=wdtmp$weights,data=wdtmp))
return(md)
}
setglm<-function(fml,yi) {
if (ydist[yi]=="") ydist[yi]<-"gaussian"
if (ydist[yi]=="exact") ydist[yi]<-"binomial"
if (ydist[yi]=="breslow") ydist[yi]<-"binomial"
if (ydist[yi]=="gaussian") md<-try(glm(formula(fml),family="gaussian",weights=wdtmp$weights,data=wdtmp))
if (ydist[yi]=="binomial") md<-try(glm(formula(fml),family="binomial",weights=wdtmp$weights,data=wdtmp))
if (ydist[yi]=="poisson") md<-try(glm(formula(fml),family="poisson",weights=wdtmp$weights,data=wdtmp))
if (ydist[yi]=="gamma") md<-try(glm(formula(fml),family="Gamma",weights=wdtmp$weights,data=wdtmp))
if (ydist[yi]=="negbin") md<-try(glm.nb(formula(fml),weights=wdtmp$weights,data=wdtmp))
return(md)
}
mdl2oo<-function(mdl, k, opt) {
gs<-summary(mdl); print(mdl$formula); print(gs)
if (opt=="gam") {gsparm <- gs$p.table;tmpn<-gs$n;
} else {gsparm <- gs$coefficients;tmpn <- sum(gs$df[c(1,2)]);}
ncolgsparm <- ncol(gsparm)
if (gs$family[[2]]=="identity") {
ncolgsparm<-5
colnm<-c("","Covariates","N","term","beta","Se.", "95%CI Low","95%CI Upp","P.value")
}
if (gs$family[[2]]=="log" | gs$family[[2]]=="logit") {
ncolgsparm<-6
colnm<-c("","Covariates","N","term","beta","Se.","exp(beta)","95%CI Low","95%CI Upp","P.value")
}
if (!smoothsvi[k]) {
if (gs$family[[2]]=="log" | gs$family[[2]]=="logit") {
gsparm<-cbind(gsparm[,c(1,2)],exp(gsparm[,1]),exp(gsparm[,1]-1.96*gsparm[,2]),exp(gsparm[,1]+1.96*gsparm[,2]),gsparm[,4])
}
if (gs$family[[2]]=="identity") {
gsparm<-cbind(gsparm[,c(1,2)],gsparm[,1]-1.96*gsparm[,2],gsparm[,1]+1.96*gsparm[,2],gsparm[,4])
}
if (substr(svname_[k],1,7)=="factor(") {
rowk<-(substr(rownames(gsparm),1,nchar(svname_[k])) == svname_[k])
} else {
rowk<-(substr(paste(rownames(gsparm),":",sep=""),1,nchar(svname_[k])+1) == paste(svname_[k],":",sep=""))
}
oo<-round(gsparm[rowk,],dec)
if (sum(rowk)==1) oo<-matrix(oo,nrow=1)
col0<-rep(k,sum(rowk))
col1<-rep(" ",sum(rowk)); col1[1]<-sb[k]
col2<-rep(" ",sum(rowk)); col2[1]<-tmpn
col3<-rownames(gsparm)[rowk]
oo<-cbind(col0,col1,col2,col3,oo)
} else {
oo <- gs$s.table[svname_[k],c(1,4)]
oo<-c(k,sb[k],tmpn,paste(svname_[k]," (edf=",round(oo[1],1),")",sep=""),rep(" ",ncolgsparm-1), round(oo[2],4))
}
if (is.vector(oo)) oo<-matrix(oo,nrow=1)
colnames(oo)<-colnm
return(oo)
}
mdl2bb<-function(mdl, j, opt) {
gs<-summary(mdl); print(mdl$formula); print(gs)
if (opt=="gam") {gsparm <- gs$p.table;} else {gsparm <- gs$coefficients;}
if (substr(xvname_[j],1,7)=="factor(") {
rowj<-(substr(rownames(gsparm),1,nchar(xvname_[j])) == xvname_[j])
} else {
rowj<-(substr(paste(rownames(gsparm),":",sep=""),1,nchar(xvname_[j])+1) == paste(xvname_[j],":",sep=""))
}
bb<-round(gsparm[rowj,1],4)
names(bb)<-rownames(gsparm)[rowj]
return(bb)
}
if (!is.na(weights.var)) {weights<-WD[,weights.var];} else {weights<-1;}
WD<-cbind(WD,weights);
vlabelN<-(substr(vlabel,1,1)==" ");
vlabelZ<-vlabel[vlabelN];vlabelV<-vlabel[!vlabelN]
vnameV<-vname[!vlabelN];vnameZ<-vname[vlabelN]
if (length(avname)>0) {df0<-as.matrix(WD[,c(xvname,avname,svname)]);
} else {df0<-as.matrix(WD[,c(xvname,svname)]);}
if (par1=="") par1<-10;
vifChk<-vifSelect(df0,t=par1,fix<-(1:length(xvname)),weights=WD$weights)
slt.vv<-vifChk[[1]];
if (sum(!slt.vv)>0) {
if (length(avname)>0) {
tmp<-slt.vv[1:length(avname)]; slt.vv<-slt.vv[-(1:length(avname))]
avname<-avname[tmp]; saf<-saf[tmp]; alv<-alv[tmp]; nadj<-length(avname)
}
svname<-svname[slt.vv]; sdf<-sdf[slt.vv]; slv<-slv[slt.vv];
}
rm(df0)
if (!is.na(subjvname)) {
if (length(avname)>0) saf<-rep(0,length(saf));
sdf<-rep(0,length(sdf));
}
fml0<-""; na=0; avb=""; cmp0=TRUE; smoothav=FALSE
if (length(avname)>0) {
na<-length(avname)
cmp0<-(apply(is.na(cbind(WD[,avname])),1,sum)==0)
avb<-vlabelV[match(avname,vnameV)];avb[is.na(avb)]<-avname[is.na(avb)]
avb[((saf=="s" | saf=="S") & alv==0)]<-paste(avb[((saf=="s" | saf=="S") & alv==0)],"(smooth)");
avname_ <- avname
smoothavi<-((saf=="s" | saf=="S") & alv==0)
smoothav = (sum(smoothavi)>0)
avname_[smoothavi]<-paste("s(",avname[smoothavi],")",sep="")
avname_[alv>0]<-paste("factor(",avname[alv>0],")",sep="")
fml0<-paste("+",paste(avname_,collapse="+"))
}
nx<-length(xvname); xb<-vlabelV[match(xvname,vnameV)]; xb[is.na(xb)]<-xvname[is.na(xb)]
ny<-length(yvname); yb<-vlabelV[match(yvname,vnameV)]; yb[is.na(yb)]<-yvname[is.na(yb)]
ns<-length(svname); sb<-vlabelV[match(svname,vnameV)]; sb[is.na(sb)]<-svname[is.na(sb)]
xvname_ <- xvname
xvname_[xlv>0]<-paste("factor(",xvname[xlv>0],")",sep="")
svname_ <- svname
smoothsvi<-((sdf=="s" | sdf=="S") & slv==0)
smoothsv <- (sum(smoothsvi)>0)
svname_[smoothsvi]<-paste("s(",svname[smoothsvi],")",sep="")
svname_[slv>0]<-paste("factor(",svname[slv>0],")",sep="")
fml1<-paste("+",paste(svname_,collapse="+"))
for (k in (1:ns)) {
tmp<-paste(svname_[-k],collapse="+"); if (tmp>" ") tmp<-paste("+", tmp)
fml1<-c(fml1, tmp)
}
if (smoothav) {gamk0<-rep(TRUE,ns+1); gamk1<-gamk0;
} else {
gamk0<-FALSE; gamk1<-smoothsv
for (k in (1:ns)) {gamk0<-c(gamk0,smoothsvi[k]); gamk1<-c(gamk1,(sum(smoothsvi[-k])>0));}
}
cmp1<-(apply(is.na(cbind(WD[,svname])),1,sum)==0)
w<-c("<!DOCTYPE html><html lang='zh'><head><meta charset='utf-8'></head><body>")
w<-c(w,paste("<h2>", title, "</h2>"))
sink(paste(ofname,".lst",sep=""))
tmp<-cbind(WD[,c(xvname,svname)]);
if (length(avname)>0) tmp<-cbind(tmp,WD[,avname])
if (sum(chkMatrixSinglar(tmp))>0) {
w<-c(w,"变量之间存在完全相关,查看.lst文件看相关系数,请重新设置")
} else {
w<-c(w,"</br>VIF 共线性筛查")
w<-c(w,"</br><table border=3>", mat2htmltable(vifChk[[3]]), "</table>")
if (length(vifChk[[2]])>0) w<-c(w,paste("</br>共线性筛查剔除的变量:",vifChk[[2]],collapse=" "))
sltoo<-c("Y","X","选出的协变量(标准1)","选出的协变量(标准2)")
for (i in (1:ny)) {
ooyi <- NA
for (k in (1:ns)) {
wdtmp<-WD[(!is.na(WD[,yvname[i]]) & !is.na(WD[,svname[k]]) & cmp0),]
fmlk<-paste(yvname[i],"~",svname_[k],fml0);
print(fmlk)
if (smoothav | smoothsvi[k]) {ootmp<-mdl2oo(setgam(fmlk,i),k, "gam");
} else {
if (!is.na(subjvname)) {
wdtmp<-wdtmp[order(wdtmp[,subjvname]),]; ootmp<-mdl2oo(setgee(fmlk,i),k, "gee")
} else {ootmp<-mdl2oo(setglm(fmlk,i),k, "glm");}
}
if (is.data.frame(ootmp)) {
for (z in (1:nrow(ootmp))) ooyi<-rbind(ooyi,as.matrix(ootmp)[z,])
} else {ooyi<-rbind(ooyi,ootmp);}
}
sltyi<-(as.numeric(ooyi[-1,ncol(ooyi)])<0.1)
sltyi<-as.numeric(unique(ooyi[-1,1][sltyi]))
slti <- rep(FALSE,ns)
slti[sltyi]<-TRUE
ooyi[,-(1:4)]<-numfmt(ooyi[,-(1:4)],4); ooyi[,ncol(ooyi)]<-pvformat(ooyi[,ncol(ooyi)],4)
ooyi[1,]<-colnames(ooyi)
w<-c(w,paste("</br></br>Y=", yb[i]))
w<-c(w, "</br></br>1. 逐个查看协变量与Y的关系")
w<-c(w,"</br><table border=3>", mat2htmltable(ooyi[,-1]), "</table>")
w<-c(w,"</br></br>2. 在基本模型中引进协变量与在完整模型中剔除协变量, 观察X的回归系数的变化")
for (j in (1:nx)) {
wdtmp<-WD[(!is.na(WD[,yvname[i]]) & !is.na(WD[,xvname[j]]) & cmp1 & cmp0),]
if (!is.na(subjvname)) wdtmp<-wdtmp[order(wdtmp[,subjvname]),]
fmlb<-paste(yvname[i],"~",xvname_[j],fml0, c("",paste("+",svname_)))
fmlc<-paste(yvname[i],"~",xvname_[j],fml0,fml1)
bb<-NA; cb<-NA
for (k in (0:ns)) {
print(fmlb[k+1])
if (gamk0[k+1]) {bb<-rbind(bb,mdl2bb(setgam(fmlb[k+1], i), j, "gam"));
} else {
if (!is.na(subjvname)) {bb<-rbind(bb,mdl2bb(setgee(fmlb[k+1], i), j, "gee"));
} else {bb<-rbind(bb,mdl2bb(setglm(fmlb[k+1], i), j, "glm"));}
}
if (gamk1[k+1]) {cb<-rbind(cb,mdl2bb(setgam(fmlc[k+1], i), j, "gam"));
} else {
if (!is.na(subjvname)) {cb<-rbind(cb,mdl2bb(setgee(fmlc[k+1], i), j, "gee"));
} else {cb<-rbind(cb,mdl2bb(setglm(fmlc[k+1], i), j, "glm"));}
}
}
tmpb<-cbind(bb,cb)
sltl<- (tmpb< matrix(rep(tmpb[2,]-abs(tmpb[2,])*0.1,nrow(tmpb)),nrow=nrow(tmpb),byrow=TRUE))
sltg<- (tmpb> matrix(rep(tmpb[2,]+abs(tmpb[2,])*0.1,nrow(tmpb)),nrow=nrow(tmpb),byrow=TRUE))
slt <- rep("",nrow(tmpb)-2)
sltij<-(apply((sltl | sltg),1,sum) > 0)[-c(1,2)]
sltijbb<-c(FALSE,FALSE,((sltl | sltg)>0)[-c(1,2),1])
sltijcb<-c(FALSE,FALSE,((sltl | sltg)>0)[-c(1,2),2])
slt[sltij]<-"Yes"
slt<-c("","",slt)
bb[1,]<-""; cb[1,]<-""; bb[1,1]<-"基本模型"; cb[1,1]<-"完整模型"
bb[-1,]<-numfmt(bb[-1,],4)
cb[-1,]<-numfmt(cb[-1,],4)
bb[sltijbb]<- paste(bb[sltijbb],"*")
cb[sltijcb]<- paste(cb[sltijcb],"*")
colnmbb<-colnames(bb);
if (substr(colnmbb,1,7)=="factor(") colnmbb<-gsub(")","=",substr(colnmbb,8,99))
row1<-c("协变量","+/- term",colnmbb,colnmbb,"选出")
ooyj<-rbind(row1,cbind(c("","",sb),c("","起始回归系数",svname_),bb,cb,slt))
ooyj<-rbind(ooyj[2,],ooyj[1,],ooyj[-c(1,2),])
w<-c(w,paste("</br>X=", xb[j]))
w<-c(w,"</br><table border=3>", mat2htmltable(ooyj), "</table>")
w<-c(w,"* 表示与起始回归系数相比变化超过 10% ")
sltooi<-c(yb[i],xb[j],paste(svname[sltij],collapse=" "))
sltooi<-c(sltooi,paste(svname[slti | sltij],collapse=" "))
sltoo<-rbind(sltoo,sltooi)
rm(sltooi,sltij)
}
rm(slti)
}
w<-c(w,"</br></br>筛选出来的协变量")
w<-c(w,"</br><table border=3>", mat2htmltable(sltoo), "</table>")
w<-c(w,"</br>注释:")
w<-c(w,"</br>1. 标准1:在基本模型中引进协变量或在完整模型中剔除协变量对X的回归系数的影响>10%")
w<-c(w,"</br>2. 标准2:标准1或协变量对Y的回归系数P值<0.1")
if (na>0) w<-c(w,"</br>3. 逐个协变量对Y的模型、基本模型、完整模型中均调整了固定要调整的变量:",paste(avb,collapse=" "))
}
sink()
w<-c(w,wd.subset)
Point 8: Could you clarify a prior what was included in the full model? Looking at your model1 and model2, it appears model 2 does not fully build on model 1 for example it leaves out sex and age? Could you please explain why? Because your fully adjusted model does not account for the age and sex, it is difficult to comment on your results as age should really be adjusted for in addition to other important confounders.
Response 8: To answer the same question as the previous one, the situation has been explained in the revised manuscript.
Point 9: “DF intake had no statistically significant effect on FBD (P = 0.18)” please add that this is from chi-square test.
Response 9: It has been added as per your request. Thank you for your correction.
Point 10: Remove this sentence: “In the adjusted regression model, in previous studies, FBD symptoms were included as a covariate due to the influence of FBD symptoms on the disease activity of AS patients”, from your results. Doesn’t belong.
Response 10: It has been added as per your request. Thank you for your correction.
Point 11: Unadjusted results have extensively been presented in the text and discussed which really is misleading. I would urge author to focus on adjusted results acknowledging that there might be residual confounding given not many factors were controlled for.
Response 11: For my previous work, I have added in the revised manuscript, detailing the adjustment of the model. The aim is to have stable results under different models to verify the reliability of the results. So it is necessary to present the unadjusted model results.
Reviewer 2 Report
Please describe clearly the methodological type of this study in the text.
Author Response
Point 1:Please describe clearly the methodological type of this study in the text.
Response 1:The type of study in this paper is a cross-sectional study, and this description has been updated in the revised manuscript; your correction is appreciated.
Reviewer 3 Report
This is a very interesting article investigating the relationship between dietary fiber intake and disease activity in ankylosing spondylitis. The study appears scientifically sound, and no further areas for improvement are readily apparent. The study is well designed with a large sample size of 165 subjects. Methods regarding data collection (dietary pattern and disease activity) and statistical analysis are explained in a detailed way such that reproducibility would be able to be assessed by other investigators. Results are thoroughly discussed and clearly presented in tabular form. Lastly, the conclusions drawn by the authors appropriately follow from the results, and a plausible mechanism by which dietary fiber modulates inflammation is provided. This manuscript therefore adequately describes the important study that was performed here, with important implications in the management of ankylosing spondylitis.
Author Response
Thank you for your kind words about the article. I am glad to hear that you found it interesting and scientifically sound. I appreciate your positive feedback on the study design, data collection, statistical analysis, and presentation of results. I am also glad that you found the conclusions drawn by the authors to be appropriate and the mechanism provided to be plausible. Thank you for taking the time to carefully review the article.